# Non-Pharmacological Approach to Diet and Exercise in Metabolic-Associated Fatty Liver Disease: Bridging the Gap between Research and Clinical Practice

**DOI:** 10.3390/jpm14010061

**Published:** 2024-01-01

**Authors:** Hassam Ali, Muhammad Shahzil, Vishali Moond, Maria Shahzad, Abhay Thandavaram, Alina Sehar, Haniya Waseem, Taha Siddiqui, Dushyant Singh Dahiya, Pratik Patel, Hans Tillmann

**Affiliations:** 1Department of Gastroenterology, Hepatology & Nutrition, ECU Health Medical Center, Brody School of Medicine, Greenville, NC 27834, USA; 2Division of Gastroenterology, Hepatology & Nutrition, East Carolina University, Greenville, NC 27834, USA; 3Department of Internal Medicine, Weiss Memorial Hospital, Chicago, IL 60640, USA; shahzilm7@gmail.com; 4Department of Internal Medicine, Saint Peter’s University Hospital, Robert Wood Johnson Medical School, New Brunswick, NJ 08901, USA; 5Department of Internal Medicine, Jacobi Medical Center, Albert Einstein College of Medicine, Bronx, NY 10461, USA; 6Department of Internal Medicine, Kamineni Academy of Medical Sciences and Research Centre, Hyderabad 500068, Telangana, India; 7Department of Internal Medicine, University of Alabama at Birmingham-Huntsville Campus, Huntsville, AL 35801, USA; 8Department of Internal Medicine, Advent Health Tampa, Tampa, FL 33613, USA; 9Department of Internal Medicine, Mather Hospital, Hofstra University Zucker School of Medicine, Port Jefferson, NY 11777, USA; tsiddiqui0095@gmail.com; 10Division of Gastroenterology, Hepatology & Motility, The University of Kansas School of Medicine, Kansas City, KS 66103, USA; 11Department of Gastroenterology, Mather Hospital, Hofstra University Zucker School of Medicine, Port Jefferson, NY 11777, USA

**Keywords:** metabolic-associated fatty liver disease (MASLD), nonalcoholic fatty liver disease, coffee, diet, ketogenic, intermittent fasting, resistance training

## Abstract

This review provides a practical and comprehensive overview of non-pharmacological interventions for metabolic-associated fatty liver disease (MASLD), focusing on dietary and exercise strategies. It highlights the effectiveness of coffee consumption, intermittent fasting, and Mediterranean and ketogenic diets in improving metabolic and liver health. The review emphasizes the importance of combining aerobic and resistance training as a critical approach to reducing liver fat and increasing insulin sensitivity. Additionally, it discusses the synergy between diet and exercise in enhancing liver parameters and the role of gut microbiota in MASLD. The paper underscores the need for a holistic, individualized approach, integrating diet, exercise, gut health, and patient motivation. It also highlights the long-term benefits and minimal risks of lifestyle interventions compared to the side effects of pharmacological and surgical options. The review calls for personalized treatment strategies, continuous patient education, and further research to optimize therapeutic outcomes in MASLD management.

## 1. Introduction

Fat accumulation in the liver that was not explained by alcohol intake was for over 30 years labeled as “nonalcoholic fatty liver disease (NAFLD)”. The term “nonalcoholic fatty liver disease (NAFLD)” was first introduced in 1986 by Schaner and Thaler [1]. However, the term “nonalcoholic” has several significant limitations. First, defining a disease entity by what it is not felt unsatisfactory. It implies complete abstinence from alcohol use, leaving a gray zone for patients who consume alcohol at moderate levels. It ignored that many people may not have both alcohol-induced and metabolic disease; some felt the term “fatty” was too stigmatizing.

In early 2020, a group of international experts led a consensus-driven effort to create a more suitable name for the disease. Through a two-stage Delphi consensus process, they proposed the term “metabolic dysfunction-associated fatty liver disease” (MASLD) [2,3]. Three years later, the term metabolic dysfunction-associated steatotic liver disease (MASLD) has been proposed, and it can be diagnosed based on a patient meeting one of five cardiovascular risk factors, unlike MAFLD, which required that patients met two of seven parameters of metabolic dysfunction. Among them, patients who meet both MASLD and alcohol-related fatty liver disease (ALD) criteria are categorized as having MetALD [4,5].

The definition of MASLD relies on the presence of hepatic steatosis and at least one other condition, such as overweight/obesity, T2DM, or metabolic abnormalities, with no additional exclusion criteria. These metabolic abnormalities include at least two of the following: increased waist circumference, low high-density cholesterol (HDL-C), hypertriglyceridemia, arterial hypertension, insulin resistance, subclinical inflammation, and prediabetes [1,2,6]. The term MASLD offers a comprehensive, non-stigmatizing portrayal of the condition and moves away from an exclusionary diagnosis. The definition now mandates the presence of at least one cardiometabolic risk factor alongside hepatic steatosis [7].

MASLD is a multifactorial disease characterized by the accumulation of excess fat (steatosis) in the liver, typically constituting 5% or more of the liver, as diagnosed through liver imaging or biopsy. It is usually associated with metabolic risk factors, such as obesity and type 2 diabetes (T2DM) while excluding excessive alcohol consumption (≥30 g per day for men and ≥20 g per day for women) and other chronic liver diseases [1,8,9]. However, MASLD without diabetes or obesity in lean individuals has been reported and is diagnosed based on BMI: <25 kg/m^2^ for non-Asians and <23 kg/m^2^ for Asians. Those with “lean MASLD” should be evaluated for comorbidities like type 2 diabetes [10]. Furthermore, even for patients with “Lean MASLD”, the first line of treatment is lifestyle optimization, including some weight loss. Although the term “steatotic liver disease” (SLD) may be less stigmatizing than “fatty liver disease”, an emerging concern is that this might shift focus from the body’s abundant fat and may complicate patient education, and needs to be evaluated in the future. 

Numerous factors are believed to play a role in the development and advancement of MASLD. These factors encompass obesity, unhealthy dietary patterns, a sedentary lifestyle, genetic and epigenetic influences, and environmental factors, as well as insulin resistance and type 2 diabetes. Collectively, these elements contribute to hepatic insulin resistance, elevated insulin levels (hyperinsulinemia), inflammation, oxidative stress, mitochondrial dysfunction, an imbalance in pro-inflammatory cytokines, and the progression from steatosis to fibrosis [11]. Most individuals with MASLD rarely exhibit any noticeable symptoms; however, some signs/symptoms may include fatigue, discomfort in the right upper quadrant of the abdomen, hepatomegaly (enlarged liver), acanthosis nigricans (a skin condition characterized by dark patches), and lipomatosis [12].

The spectrum of MASLD spans from mild steatosis to steatohepatitis, progressing to fibrosis and cirrhosis and eventual decompensation and hepatocellular carcinoma (HCC) [12,13,14]. Not only is type 2 diabetes mellitus a risk factor for MASLD, but MASLD is also a risk factor for developing type 2 diabetes cardiovascular disease (CVD) and kidney disease [15]. The general risk of non-liver-related cancer seems to be increased with MASLD, which may be related to insulin resistance. Importantly, MASLD serves as an independent risk factor for both liver-related and overall mortality, underscoring the urgency of halting the progression of this condition [11].

Lifestyle changes offer a promising therapeutic strategy for addressing MASLD. Weight loss of at least 10% has been linked to the resolution of steatohepatitis and even the reversal of fibrosis by at least one stage in some cases [11,13]. Even modest weight loss (over 5%) has benefits in several aspects assessed in the MASLD activity score. For instance, a 5% decrease in BMI has been associated with a 25% reduction in liver fat, as measured by magnetic resonance imaging (MRI) [16]. Complete correction of mildly elevated LFTs can sometimes be achieved after a few weeks on a strictly hypocaloric diet. Dietary interventions have been shown to improve MASLD, both with and without physical activity [16]. However, there is ongoing debate regarding the specific composition of the diet and dietary patterns. In contrast, when it comes to exercise interventions, there is consensus in the literature that various exercise modalities and intensities are generally beneficial for MASLD [17,18]. Exercise has consistently been demonstrated to reduce hepatic steatosis and lower liver enzyme levels. This likely is the result of improved blood glucose control, enhanced insulin sensitivity, and improved lipid profiles, whether employed alone or with dietary changes [19].

This review aims to link the research evidence to practical applications in the management of MASLD. We will explore the latest insights into the role of diet, exercise, pharmacology, and endoscopic/surgical interventions in preventing, facilitating, and potentially reversing MASLD. 

## 2. Epidemiology

MASLD has emerged as the most prevalent chronic liver disease worldwide in the last 20 years, affecting an estimated 25% of the global population [3,15]. A meta-analysis by Liu et al. estimated the global prevalence of MASLD in overweight/obese adults to be approximately 50.7% (95% CI [46.9% to 54.4%]) [20]. Current estimates show that MASLD affects 30% of the United States (US) population, 32% of the Middle Eastern population, 30% of the South American population, 27% of Asian populations (highest in East Asians), 24% of the European population, and 13% of the African population (Figure 1) [8,9]. In the US, Hispanic Americans (45%) have a higher prevalence of MASLD than Caucasians (33%), while African Americans (24%) have the lowest prevalence among all racial and ethnic groups (Figure 1) [21]. Furthermore, among the Hispanic population, those of Mexican heritage have the highest prevalence, while Dominican Republicans have the lowest prevalence [8,22]. In the US, men are disproportionately affected, making up to 58% of total MASLD cases [23]. At the same time, postmenopausal women display an increased risk of severe fibrosis compared to men, which can probably partially be attributed to the loss of the protective effects of estrogen against fibrogenesis [13]. However, why the loss of a protective factor results in a more accelerated deterioration in men remains to be elucidated. One potential explanation may be the lower muscle mass at similar BMIs.

The prevalence of MASLD in children is 3–10%, but it can increase to 40–70% among obese children [24]. In developed countries, MASLD affects 16.9–23.8% of boys and 16.2–22.6% of girls, while in developing countries, the prevalence is 8–12% in boys and 8–13% in girls [25]. It is diagnosed in 47.3–63.7% of individuals with T2DM and up to 80% of those with obesity [26,27]. 

In the United States, the population affected by MASLD is anticipated to increase from 83.1 million in 2015 (approximately 25% of the population) to 100.9 million by 2030 [22], which is more than 1 in 4 Americans (Figure 2). This scale also indicates that non-pharmacological interventions will be important, as it is not sustainable to treat all patients pharmacologically, specifically at current prices.

Although MASLD is often associated with central obesity in North America and Europe, affecting roughly 83% of obese patients, it is noteworthy that in Asia, a significant percentage of individuals with MASLD have a body mass index (BMI) below [25], a condition referred to as lean-NAFLD/MASLD [28,29,30]. It is essential to recognize that the BMI cutoff for defining overweight in Asia (BMI > 23) is lower than that in North America and Europe (BMI > 25) [19,22]. Notably, cirrhosis secondary to MASLD is the second leading cause of liver transplantation in the Western world [1,9,13]. 

## 3. Risk Factors

Several risk factors have been identified, which will be reviewed individually below. 

### 3.1. Metabolic Syndrome

Metabolic syndrome (MetS) is a prominent and significant risk factor for developing MASLD. MetS exhibits variable definitions but typically encompasses increased waist circumference (a better indicator of obesity than BMI), hyperglycemia, dyslipidemia, and systemic hypertension (HTN) [22,31]. The clustering of disease comorbidities manifesting as MetS is by far the most characteristic feature of NAFLD, present in 36–67% of patients [32]. The relationship between MASLD and MetS features can be bidirectional, especially with diabetes and HTN. In essence, not only does MetS heighten the risk of MASLD, but MASLD may also exacerbate several features and comorbidities associated with MetS [22,27]. 

### 3.2. Type 2 Diabetes Mellitus

MASLD prevalence is high in individuals with impaired glucose tolerance and newly diagnosed diabetes at the proportion of 43% to 62%, respectively [33]. In a prospective study of 100 patients with type 2 diabetes, the incidence of hepatic steatosis was 49%, suggesting this strong independent risk factor for MASLD [34]. Diabetes mellitus is the most evident biological connection to the development of MASLD, with as many as 75% of individuals with type 2 diabetes also having MASLD [35]. Diabetes is not only a common comorbidity of MASLD but also one of the determinants of the progression of MASLD toward MASH, developing accelerated liver fibrosis and HCC [36]. There is a higher prevalence of steatohepatitis and advanced fibrosis in individuals with both diabetes and MASLD [37]. Type 2 diabetes promotes MASLD progression to cirrhosis and elevates the risks of liver-related and all-cause mortality by two- to three-fold. In a recent study involving 713 participants with biopsy-proven MASLD (48% with type 2 diabetes), it was shown that each 1% increase in mean glycated hemoglobin (HbA1c) in the year preceding liver biopsy was independently associated with a 15% higher odds of harboring more severe stages of liver fibrosis, highlighting the effect of glycemia on fibrosis progression [38]. The relation between MASLD and NIDDM is bidirectional, with each contributing to the development and progression of the other disease [39]. In large studies of individuals without diabetes at inclusion, the presence of MASLD was associated with an increased adjusted risk of 2.4–3.5 for developing diabetes [40].

### 3.3. Hypertension

Hypertension, especially systolic hypertension, is also an independent MASLD predictor [34]. A pioneering cohort study enrolling 1635 Nagasaki atomic bomb survivors without MASLD at baseline reported that HTN significantly predicted the development of ultrasound-diagnosed MASLD over a mean follow-up of 11.6 years [41,42]. Another study examining the existence of a reciprocal causality between MASLD and MetS has confirmed that MetS components predict the risk of incident MASLD, although with some variability [41,43]. The Framingham Heart Study investigators showed that HTN and other MetS components were all similarly associated with increased odds of incident MASLD, independently of adiposity measures, over approximately six years of follow-up [41,44]. All these findings clearly suggest that HTN and probably pre-HTN play a role in MASLD development. However, HTN appears to have a smaller effect on the risk of incident MASLD than other MetS components [41,45]. In another study, 50% of patients with HTN have MASLD [22], and MASLD has been associated with changes in arterial stiffness, myocardial remodeling, kidney disease, and heart failure. HTN is strongly associated with fibrosis progression [22].

### 3.4. Obesity

Obesity is nearly ubiquitous among patients with MASLD, as up to 75% of patients who are overweight and 90–95% of patients with morbid obesity have MASLD. The distribution of adiposity, specifically the presence of truncal obesity, is a more important determinant of MASLD risk than body mass index (BMI) [32]. Although obesity is intimately associated with liver fat, not all patients with obesity develop metabolic fatty liver disease. Current consensus suggests that the distribution and the overall health of fat, rather than its amount, is likely the major determinant of disease risk. For example, higher amounts of visceral relative to peripheral and subcutaneous adipose tissue are associated with greater metabolic risk and are directly linked to liver inflammation and fibrosis, independent of insulin resistance and hepatic steatosis [46]. In MASLD, a normal BMI but an obese waist circumference is associated with a higher risk of mortality secondary to cardiovascular disease compared to those who are overweight by BMI but have an average waist circumference [15]. For waist circumference, measurements greater than 102 cm in Caucasian men and greater than 88 cm in Caucasian women, or greater than 90 cm in Asian men and greater than 80 cm in Asian women, are often used as thresholds to assess abdominal obesity [2]. Increased visceral adipose tissue leads to, among numerous other deleterious effects, insulin resistance and subsequent hyperinsulinemia, which increases circulating FFA, hepatic VLDL production, and gluconeogenesis. Visceral adiposity also increases portal blood FFA that drains into the liver. The combination of these mechanisms results in excessive hepatic steatosis. Progressive steatosis may result in hepatocellular lipotoxicity through cellular and organelle oxidative stress; endoplasmic reticulum stress and mitochondrial dysfunction drive reactive oxygen species production, leading to hepatic ballooning, inflammation, and cell death through apoptosis, histological hallmarks of MASH [47]. Obesity is associated with increased adipose tissue lipolysis and secretion of inflammatory/fibrotic mediators that can reach the liver. In adipose tissue, the accumulation of inflammatory and immune cells and alterations in their activities give rise to chronic low-grade inflammation [13,48]. Ongoing inflammation plays a role in mediating insulin resistance and the subsequent development of MASLD [13]. Obesity also causes the secretion of adipokines (e.g., leptin, adiponectin) and hormones in the liver, which can contribute to the progression of MASLD to MASH cirrhosis and HCC [36].

### 3.5. High Fructose Intake

Unlike glucose, fructose is predominantly metabolized in the liver through the action of glucose transporter type-5 (GLUT 5). This metabolic pathway leads to the significant production of acetyl-CoA after fructose uptake, bypassing glycolysis, the rate-limiting step in acetyl-CoA generation [49]. Although some acetyl-CoA is utilized for ATP production, the surplus is directed toward de novo lipogenesis, a proposed mechanism contributing to the development of MASLD when fructose is consumed in excess [49]. Beyond its role in lipogenesis, fructose also exerts hepatotoxic effects by directly generating reactive oxygen species (ROS), contributing to hepatocellular damage through protein fructosylation [26]. 

### 3.6. Gut Microbiota

The gut–liver axis signifies the connection between gut microbiota and the liver, facilitated by the portal vein, which transports substances from the gut to the liver and reciprocally carries bile and antibodies from the liver to the intestine [50]. A crucial determinant of health appears to be the mucosal barrier, formed by intestinal epithelial cells, with its permeability and mucus composition influenced by gut microbiota and immune cell presence. The elevated permeability of this intestinal mucosal barrier and unfavorable alterations in gut microbiota composition are potential contributors to the onset and progression of MASLD [50]. Dysfunctional gut microbiota leads to the generation of PAMPs (pathogen-associated molecular patterns), while increased mucosal barrier permeability results in heightened liver inflammation, contributing to the development and progression of liver disease [1,13].

### 3.7. Genetics

Presently, at least five distinct genetic variants are strongly linked with an increased susceptibility to MASLD progression. These variants are in PNPLA3, TM6SF2, GCKR, MBOAT7, and HSD17B13 genes [3]. Some of these genetic variations also elevate the risk of developing T2DM, such as TM6SF2, TCFL2, and SREBF2, while others are associated with obesity risk, like ADIPOQ and SH2B1 [3]. As an example, some of these variables likely explain the increased risk of MASH and the related increased prevalence of fibrosis among Hispanic Americans. Although previously all Hispanics were thought to be at risk, more recent data suggest that origin also plays a role, wherein Hispanics of Mexican origin have a prevalence rate of 33% and those of Puerto Rican and Dominican descent have prevalence rates of 18% and 16%, respectively. These differences are attributable, in part, to the difference in carriage of a single polymorphism in the PNPLA3 gene, which has a higher relative frequency in Mexico. Other countries, such as Japan and South Korea, have a higher relative frequency of the PNPLA3 risk allele [32]. Although African Americans have high rates of metabolic syndrome, MASLD, particularly MASH, is less common in African Americans compared with Caucasian and Hispanic populations. This is attributable, in part, to lower carriage rates of PNPLA3; however, differences in adipose tissue distribution also play a role [32]. These genes play roles in insulin resistance, glucose regulation, and lipid balance, suggesting shared mechanisms between MASLD, T2DM, and obesity in their pathogenesis [3]. The most well-known is a single-nucleotide polymorphism in the PNPLA3 gene, responsible for regulating hepatocyte lipid droplet breakdown. This genetic variant, PNPLA3-I148M, is strongly linked to steatohepatitis and resists expected degradation, accumulating on lipid droplets and impeding lipolysis. Notably, the risk of steatohepatitis associated with this variant is highest when coupled with excess body fat, highlighting the combined impact of genetic and environmental factors on the disease [22,51]. Although these genetic advancements have increased our understanding of the pathogenesis of MASLD and may account in some part for the so-called “lean” MASLD patients, testing for these genetic variants is currently still not advocated in routine clinical practice [32], as none would currently change management of the disease. This may change in the future if the effectiveness of some treatments depends on underlying genetics. For example, IL28B is an interferon-based therapy for hepatitis C [52] but not for hepatitis B [53], where the HBV genotype plays a role [54].

### 3.8. Aging

Recent literature reinforces the notion that aging, particularly biological age, plays a pivotal role in the progression of metabolic-associated fatty liver disease (MASLD) [55]. Older age is seen to potentially accelerate the rate of MASLD progression, a phenomenon likely linked to the progressive tissue degeneration that accompanies the aging process. This effect is more profoundly influenced by biological age rather than chronological age, underscoring the significance of epigenetic factors [56]. Epigenetics involves a dynamic maintenance system that regulates gene activity (phenotype) without altering the DNA code itself. The interaction between aging and epigenetics is crucial in understanding the progression of MASLD. It highlights how age-related changes in the epigenome, often manifested as altered DNA methylation patterns and epigenetic noise, contribute significantly to disease severity. These epigenetic alterations not only reflect the cumulative impact of an individual’s lifestyle and environmental exposures but also determine how effectively the body can respond to the stresses imposed by aging and disease processes.

## 4. Pathophysiology of MASLD

MASLD has various proposed pathophysiological mechanisms described, with four main mechanisms that contribute to the accumulation of triglycerides in the liver, including de novo lipogenesis, increase in the uptake of fatty acids, triglyceride secretion, and oxidation of fatty acids (Figure 3).

### 4.1. Hepatic De Novo Lipogenesis (DNL)

De Novo Lipogenesis (DNL) is a process where our cells turn extra carbohydrates, mainly glucose, into fatty acids [19] (Figure 1). This process is especially important when we talk about obesity and MASLD. Studies have shown that in obese people with MASLD, DNL is responsible for about 20–30% of the fat that accumulates in the liver [57,58]. One key reason for this is insulin resistance, often seen in obesity. Insulin usually helps to control how much fat is broken down in our bodies. However, when insulin resistance occurs, this control is lost, leading to more fat being released into the bloodstream [14]. This excess fat ends up being stored in the liver, contributing to MASLD [9,59]. Interestingly, not all carbohydrates have the same impact on DNL. Fructose, for example, is more effective, and therefore deleterious, than glucose at triggering this fat-making process [60,61]. Besides carbohydrates, amino acids, and short-chain fatty acids also contribute to DNL. Three enzymes—acetyl-CoA carboxylase (ACC), fatty acid synthase (FAS), and stearoyl-CoA desaturase-1 (SCD1)—are crucial for this process. ACC starts by converting one molecule into another, which FAS then turns into a type of fatty acid. SCD1 adds the finishing touches by modifying these fatty acids [48]. These enzymes are controlled by “master switches” called SREBP-1c and ChREBP [9,48]. When insulin resistance happens, it activates SREBP-1c, which in turn ramps up DNL in liver cells [62] (Figure 4).

### 4.2. Fatty Acid Oxidation

Fatty acid oxidation (FAO) in cells generally matches plasma-free fatty acids (FFAs) levels [63]. FFAs, acquired from the bloodstream or released through cellular processes, are mainly oxidized in mitochondria, peroxisomes, and microsomes, primarily handling very long-chain, branched-chain, and unsaturated fatty acids [14]. The status of hepatic FAO in MASLD remains uncertain due to contradictory findings and a lack of in vivo measurement of mitochondrial oxidation. Indirect indicators, such as plasma ketone body concentrations, suggest that FAO in the liver may either increase or decrease in MASLD. Significantly, the liver tissue in patients with steatotic liver disease unequivocally exhibits decreased activities in all mitochondrial respiratory chain complexes [60].

The transport of fatty acids into mitochondria is contingent on carnitine palmitoyltransferase 1 (CPT1), an enzyme located in the outer mitochondrial membrane. A key controller of CPT1 activity is the peroxisome proliferator-activated receptor-α (PPARα). When PPARα is activated, it stimulates the transcription of genes involved in fatty acid oxidation within mitochondria (such as CPT1), peroxisomes (including acyl-CoA oxidase or ACOX), and cytochromes (like the Cyp4A family) [48]. When peroxisomal enzymes (such as acyl-coenzyme A oxidase or peroxisomal biogenesis factor 11α) are deficient, peroxisomal β-oxidation is impaired, increasing lipid accumulation and exacerbating hepatocyte damage [48]. This highlights the crucial role of peroxisomes in lipid metabolism in the context of MASLD. In the case of microsomal ω-oxidation, an excess of hepatic fatty acids serves as both a substrate and an inducer of cytochrome P-450 enzymes in microsomes. Hence, these organelles and mitochondria play a significant role in generating reactive oxygen species (ROS) [14].

### 4.3. VLDL Excretion

VLDL consists of a triglyceride-rich core surrounded by phospholipids and proteins like apolipoprotein B-100, aiding in its function. Interestingly, most of the triglycerides in VLDL come from external sources, not de novo lipogenesis. In patients with steatohepatitis, impaired apolipoprotein B-100 synthesis may be linked to elevated free fatty acids, disrupted redox balance, hyperinsulinemia, and reduced expression, leading to liver lipid accumulation [14].

In MASLD, there is an increase in the release of triglyceride-enriched lipoproteins from the liver; however, there is much higher fat buildup within hepatocytes. Patients with steatohepatitis, in particular, experience reduced synthesis and export of lipoproteins, implying a limited ability to remove VLDL particles, possibly due to hepatocyte impairment [14]. Additionally, patients with MASLD produce increased small, dense LDL particles, which are associated with an increased risk of cardiovascular disease.

### 4.4. Role of Insulin Resistance

Insulin resistance denotes an impaired response of target tissues when exposed to insulin. The underlying cause of this anomaly is multifactorial, stemming from various factors that collectively contribute to this metabolic abnormality. These factors include alterations in the number or malfunction of insulin receptors, deficiencies in the insulin signaling pathway, suppression of the pathway by inhibitors, and the presence of a pro-inflammatory internal environment [14]. There is also a hypothesis suggesting that fatty acids may induce IR by promoting abnormal lipid accumulation in muscles and the liver due to altered metabolism [31].

Insulin resistance plays a pivotal role in the pathophysiology of MASLD. Under normal circumstances, pancreatic beta cells release insulin primarily in response to circulating glucose levels. Insulin exerts its effects on various metabolic tissues, such as adipose tissue, by promoting the esterification of fatty acids and their storage into lipid droplets while simultaneously inhibiting the process of lipolysis [8,22]. Within hepatocytes, insulin has three primary functions: promoting glycogen storage, inhibiting gluconeogenesis, and activating key regulators of de novo lipogenesis. In individuals with MASLD, the development of insulin resistance leads to several consequences: (1) heightened adipocyte lipolysis, resulting in increased circulation of free fatty acids available for subsequent hepatic uptake; (2) reduced hepatic glycogen storage; and (3) an elevated level of gluconeogenesis [8,64].

When insulin resistance occurs, it triggers the inhibition of lipolysis in adipose tissue, leading to an increased breakdown of stored triglycerides within AT and a higher release of FFAs into the bloodstream. These circulating FFAs subsequently activate the pro-inflammatory nuclear factor kappa-light-chain-enhancer of activated B cells (NF-κB) pathway in the liver, ultimately resulting in lipotoxicity.

### 4.5. Inflammatory Pathways

Liver inflammation is governed by the equilibrium between pro-inflammatory M1 Kupffer cells (KCs) and anti-inflammatory M2 KCs. The liver is constantly exposed to various substances, such as nutrients and products from gut bacteria, which are transported through the portal circulation and cleared by KCs [19]. A prolonged high-fat diet can increase the number of KCs displaying a pro-inflammatory M1 phenotype [27]. KCs are responsible for producing a range of inflammatory cytokines, including TNF-α, IL-1β, IL-6, IL-12, and IL-18, and chemokines [19,27]. Saturated fatty acids have been observed to encourage the M1 polarization of KCs, whereas omega-3 polyunsaturated fatty acids (PUFAs) shift KCs towards the M2 phenotype. This shift was linked to the activation of the NF-κB and PPAR-γ signaling pathways, respectively. Steatohepatitis is distinguished by the significant enlargement and clustering of KCs in the perivenular regions, accompanied by the presence of scattered large fat vacuoles within these KCs [27].

Inflammation is incited through two primary pathways: the IκB kinase-b (IKKb) pathway and the JUN N-terminal kinase (JNK) pathway. Individuals with T2DM typically exhibit elevated levels of pro-inflammatory molecules in their serum, including interleukin-1b (IL-1b), IL-6, tumor necrosis factor (TNF)-alpha, and C-reactive protein (CRP), compared to those without T2DM. These inflammatory mediators activate the IKKb and JNK pathways, exacerbating insulin resistance. The downstream transcription factor nuclear factor-kB (NF-kB) further amplifies the expression of these pro-inflammatory cytokines [3].

### 4.6. Lipotoxicity

In MASLD, excess FFAs overwhelm the liver’s metabolic capacity, impairing beta-oxidation and mitochondrial dysfunction. These surplus FFAs can be converted into triglycerides, stored as lipid droplets, and partly released into the bloodstream as very low-density lipoproteins. Excessive FFAs can serve as substrates for generating lipotoxic lipid molecules such as ceramides and diacylglycerols. These lipotoxins induce stress in hepatocytes and, along with the free pool of hepatic fatty acids, contribute to mitochondrial dysfunction and endoplasmic reticulum stress. They also activate NADPH oxidase (NOX), an enzyme complex that generates superoxide free radicals, a major source of cellular ROS. These processes lead to increased production of ROS, including O^2−^, H_2_O_2_, malondialdehyde (MDA), and 4-hydroxy-2-nonenal (4-HNE). Elevated ROS levels modify insulin and innate immune signaling and affect the expression and activity of crucial enzymes involved in lipid regulation. Together, these effects result in redox-dependent dysregulation of hepatic lipid metabolism observed in MASLD. Excessive and dysregulated ROS production within the mitochondrial matrix may damage constituent structures, including the mitochondrial membrane mitochondrial DNA (mtDNA). It may induce pro-apoptotic pathways, including mitochondrial autophagy, a process also known as mitophagy [65].

It is important to note that lipotoxicity does not arise from a single pathway but rather from the combined effects of FFAs, triglycerides (TGs), bile acids (BAs), free cholesterol, ceramides, and lysophosphatidylcholines [31]. Lipotoxicity further impairs insulin signaling, causes oxidative damage, and promotes inflammation and fibrosis through several mechanisms. These downstream effects are thought to be responsible for the progression from steatosis to steatohepatitis and the development of fibrosis and hepatocellular carcinoma in MASLD [8]. Phosphorylation of c-Jun N-terminal kinases (JNKs) in adipocytes disrupts post-receptor insulin signaling, particularly during inflammation. Consequently, inflammatory and metabolic processes within adipose tissue can contribute to the development of steatohepatitis, making them potential targets for therapy [22].

### 4.7. Endoplasmic Reticulum (ER) Stress

ER stress initiates a protective response aimed at restoring protein balance by triggering the unfolded protein response (UPR) pathway through three transmembrane sensor proteins: inositol-requiring enzyme 1a (IRE1a), double-stranded RNA-dependent protein kinase (PKR)-like ER kinase (PERK) and activating transcription factor 6 (ATF6) [66]. Under typical circumstances, these molecules are bound to glucose-regulated protein 78 (GRP78), maintaining them inactive. However, when ER stress occurs, these pathways can become active as they dissociate from GRP78, influencing various downstream events that ultimately result in cell death. The ER membrane primarily comprises a limited amount of cholesterol and complex sphingolipids. This loose arrangement of lipids in the ER membrane facilitates the production of new lipids and the transport of proteins. ER stress mainly impacts the metabolic pathway of lipogenesis [9].

ER stress induces apoptosis through three sensor dimers and autophosphorylation. The phosphorylation of eukaryotic initiation factor 2α (eIF2α) by PERK temporarily reduces translation but activates transcription factor 4 (ATF4) for selective translation. ATF4 functions as a transcription factor and promotes the expression of CCAAT-enhancer-binding homologous protein (CHOP) associated with apoptosis. CHOP also serves as a substrate for ATF6. Additionally, ATF6 upregulates the X-box binding protein-1 (XBP1) expression, which mediates inflammatory responses through the JNK signaling pathway. IRE1 activates tumor necrosis factor (TNF) receptor-associated factor 2 (TRAF2) and JNK, promoting apoptosis. Multiple studies have shown that the IRE1 pathway can activate JNK via its kinase structural domain, increasing pro-inflammatory mediators’ expression. Activation of IRE1 results in the splicing of XBP1, a key transcription factor regulating genes involved in adaptive UPR. These findings suggest that ER stress contributes to the progression of steatosis to steatohepatitis [9,66].

### 4.8. Inflammasome

Inflammasomes are cytoplasmic multi-protein complexes triggered by external pathogen-associated molecular patterns (PAMPs) such as lipopolysaccharides and internal host damage-associated molecular patterns (DAMPs). DAMPs, released when hepatocytes die, can enhance inflammation by activating pattern recognition receptors on KCs and attracting inflammatory cells such as monocytes and neutrophils [59,66]. Recruited monocytes, known as LY6Chi monocytes, express high levels of lymphocyte antigen 6C. These monocytes differentiate into M1-type macrophages, which contribute to inflammation by releasing cytokines and ROS. Certain cytokines, such as transforming growth factor-b, can also directly affect hepatic stellate cells (HSCs). This can lead to the transformation of HSCs into myofibroblasts, promoting fibrogenesis. Various types of macrophages are involved in the resolution of inflammation. These are often referred to as alternatively activated macrophages or M2-type macrophages. They can counteract the pro-inflammatory environment and contribute to tissue repair. However, they can also promote fibrotic changes in the liver that are part of the development of steatohepatitis [59]. The ligand membrane receptors activate a complex intracellular cascade that produces many cytokines, including IL-18 and IL-1β, which have pro-inflammatory and profibrotic effects. The involvement of TLR2, 4, and 9 has been shown in the MASLD/steatohepatitis model. IL-1β, IL-1α, and IL-33 are pivotal in initiating the sequence of events that lead to steatohepatitis. IL-1β has been shown to stimulate the production of TNF and IL-6 and promote the accumulation of triglycerides and cholesterol by upregulating genes involved in their synthesis, such as diacylglycerol-O-acyltransferase and PPARγ. Interestingly, IL-33 administration has ameliorated steatosis but exacerbated certain aspects of steatohepatitis, including fibrosis. Additionally, individuals with fibrosis have demonstrated elevated transcript levels of IL-33 in the liver [66].

### 4.9. Mitochondrial Dysfunction

Mitochondria play a pivotal role in the regulation of hepatic lipid metabolism and the management of oxidative stress. In liver tissues of patients with both alcohol-related and non-related liver disorders, there are observable changes such as ultrastructural damage to mitochondria, altered mitochondrial dynamics, reduced respiratory chain complex activities, and a compromised capacity for adenosine triphosphate synthesis [67]. The balance shifts towards increased lipogenesis and reduced fatty acid β-oxidation, resulting in triglyceride accumulation within hepatocytes. This imbalance, along with elevated reactive oxygen species, contributes to insulin resistance in steatohepatitis patients. Mitochondrial reactive oxygen species are also key in signaling metabolic pathways and any disruption in these pathways can influence the onset and progression of chronic liver diseases. The stress and damage to mitochondria are implicated in cellular death, fibrogenesis of the liver, inflammation, and innate immune responses to viral infections. Therefore, mitochondrial functions are entwined with the development of various chronic liver conditions, such as nonalcoholic fatty liver disease, alcohol-associated liver disease, drug-induced liver injury, and viral hepatitis B and C. This exposure hints at the potential therapeutic strategies targeting these mitochondrial processes [67]. The equilibrium between the oxidation and storage of fat hinges on the type of fuel mitochondria utilize rather than their capacity to generate ATP through respiration. Therefore, therapeutic approaches that modulate mitochondrial fuel choice could be more effective for managing nonalcoholic fatty liver disease (NAFLD). In parallel, it may be more beneficial to inhibit maladaptive antioxidant responses instead of disrupting the normal mitochondrial hydrogen peroxide-driven signaling pathways, preserving proper hepatic insulin signaling in NAFLD. Investigating the specific roles of different antioxidant systems within subcellular compartments, as well as the distinct roles played by various mitochondrial subpopulations, could unveil novel targets for NAFLD treatment [68].

## 5. Importance of Diet in MASLD

Although no medications are currently approved specifically for the treatment of metabolic-associated fatty liver disease (MASLD) and related cirrhosis, lifestyle modifications, including diet and physical activity, are widely accepted as foundational treatments for NAFLD/NASH [69]. Despite being recognized as essential for addressing the NAFLD epidemic, existing guidelines offer imprecise and broad directives for dietary and exercise interventions for affected individuals [70].

Several scientific associations (EASL-EASD–EASO 2016 [71], AASLD 2018 [72], ESPEN 2019 [73], and APASL 2020 [74]) emphasize the significance of weight loss—aiming for a 7–10% reduction in body weight achieved by a hypocaloric diet (energy deficit of 500–1000 kcal/d) and/or PA (in order to promote a caloric deficit). Despite the consensus on the objective of weight loss, the specifics of the recommendations vary across different associations. 

Weight loss is crucial in reversing MASLD. We discuss different dietary approaches that target weight loss and improve liver health through a variety of mechanisms.

### 5.1. Intermittent Fasting

Intermittent fasting, a dietary approach alternating between fasting and fed states, has demonstrated notable metabolic and health benefits, including insulin sensitivity [75], weight loss, dyslipidemia, and improvements in hepatic health [76,77]. Common modes include the 5:2 diet, moderate alternate-day fasting, time-restricted feeding, and religious fasting practices [78]. Literature reviews and a recent pooled analysis with intermittent fasting as the intervention suggests a significant decrease in body weight, BMI, waist-to-hip ratio, total serum triglycerides, serum LDL cholesterol, and Homa–IR (a marker of insulin resistance). Additionally, marked enhancements in liver health were evident through improvements in liver enzymes, liver steatosis, and liver stiffness [78]. The outcomes of some of these studies highlight the potential of intermittent fasting as a potential lifestyle modification that may be of benefit in MASLD.

### 5.2. Mediterranean Diet

The Mediterranean diet primarily consists of plant-based foods such as fruit, vegetables, whole grains, legumes, seeds, nuts, and olive oil and advocates moderate consumption of fish, dairy products, alcohol, and red meat limited to two servings per month [79,80]. This diet yields significant positive effects, including reductions in total cholesterol levels, liver stiffness, and waist circumference [79,80]. Individual studies have also demonstrated benefits such as reduced BMI, weight loss, improved liver enzyme profiles, and enhanced insulin sensitivity [81,82]. Even though the recent meta-analysis of the relevant trials found a reduction in liver stiffness and total cholesterol levels, there were no statistically significant differences in improvement in liver enzymes and waist circumference between the intervention group (on the Mediterranean diet) and the control group [83]. High saturated fat, low fiber, and carbohydrate-rich diets have all been associated with MASLD risk, but little direct evidence exists in humans [8].

### 5.3. Low-Carb and Low-Fat Diet

A very low-carbohydrate or ketogenic diet typically defines a carbohydrate intake ratio to a total daily calorie intake of less than 10%. A 0–26% ratio is termed as low-carbohydrate, and 26–45% is moderate carbohydrate intake. Importantly, a low carbohydrate intake correlates with higher dietary fats. A 2019 pooled analysis comparing low-carb diets to low-fat diets revealed significant health marker improvements, including reductions in hepatic fat content (HFC), triglycerides (TG), high-density lipoprotein (HDL) levels, HbA1c, and HOMA–IR [84]. However, no significant improvements in BMI, blood pressure, fasting blood glucose (FBG), or C-reactive protein (CRP) levels were observed. A recent meta-analysis pooling five studies failed to demonstrate differences in H-MRS measurements, weight loss, intrahepatic fat levels, or liver enzyme levels such as AST and ALT [85]. Nonetheless, individual studies showed significant effects of both diets on MASLD compared to a standard diet [86]. When comparing solely low-carbohydrate diet interventions against normal diets, reductions in liver fat content were observed [87]. Subgroup analyses further unveiled favorable effects of low-carb diets, especially regarding AST levels and liver fat content. Additionally, Chai et al. indicate that low-carb diets were associated with improvements in hepatic fat content, triglyceride levels, HDL levels, HbA1c, and HOMA–IR compared to low-fat diets [88,89].

### 5.4. Ketogenic Diet

MASLD is associated with disrupted lipid metabolism, often due to mitochondrial dysfunction, which initiates a harmful cycle that exacerbates oxidative stress and triggers inflammation, leading to the progressive loss of hepatocytes and advancing MASLD to its more severe stages. A ketogenic diet, characterized by very low carbohydrate intake (less than 30 g per day) that leads to “physiological ketosis,” has shown promise in mitigating oxidative stress and improving mitochondrial function [90]. There has been evidence provided by studies that show that the ketogenic diet (KD) can enhance metabolic health and increase the population of γδ T cells within adipose tissue, which plays a critical role in controlling blood sugar levels in the context of obesity. Consequently, the KD is being considered to be a potential treatment for individuals with sarcopenic obesity due to its beneficial impacts on visceral adipose tissue (VAT), adipose tissue regulation, inflammatory markers including cytokines, blood biochemistry, gut microbiota, and overall body composition [91].

Studies on obese patients associate this diet with significant improvements, including reductions in AST, ALT, GGT levels, blood pressure, and changes in the lipid profile [92]. One meta-analysis in patients with MASLD revealed a substantial reduction in visceral adipose tissue and liver steatosis with the very low-calorie ketogenic diet [92]. Other studies have noted a significant reduction in liver fat within the ketogenic diet intervention group [93,94,95]. Cunha et al. also found that the very low-calorie ketogenic diet significantly decreased residual adipose tissue and liver steatosis compared to a low-calorie diet at a 2-month follow-up [96]. 

### 5.5. Caffeine Consumption

The mechanism through which caffeine exerts its protective effect is believed to be its capacity to curb the accumulation of fat and collagen in the liver, which are key indicators of liver disease progression. By diminishing liver inflammation and decelerating fibrosis, caffeine from coffee serves to manage MASLD and significantly contributes to overall liver health [97]. This positions regular coffee consumption as an advantageous dietary choice for MASLD patients. Interestingly, there is no significant association between total caffeine consumption and the prevalence of hepatic fibrosis of MASLD [97]. However, regular coffee, distinct from other sources of caffeine like espresso or tea, has been specifically associated with reductions in liver enzymes like GGT and ALT, signifying improved liver function. These findings align with the broader notion that coffee, in its regular form, offers unique benefits in the context of liver health, making it a valuable addition to dietary strategies for managing MASLD.

## 6. Importance of Exercise in MASLD

Although there are few emerging drugs in play for MASLD treatment, such as GLP-1 agonists [98]. The widely adapted first-line treatment for MASLD remains weight loss through portion control and exercise training [99]. In patients with MASLD, exercise has been shown to dramatically reduce ALT, insulin resistance, and the grade of liver steatosis [100,101,102].

Diminished muscle mass is linked with lower survival rates, prolonged hospital stays, and increased mortality in cirrhotic patients [103]. Muscle function also has a reciprocal relationship with MASLD. Studies have shown that individuals with reduced muscle mass are at a heightened risk of developing MASLD, even when factors like insulin resistance (IR) and inflammation are accounted for [104]. Skeletal muscle index (SMI) has a converse relationship with markers such as HOMA–IR, hs-CRP, triglycerides, and overall body fat percentage MASLD even after adjusting for potential confounding factors [104]. Another subsequent study showed a positive relationship between sarcopenia and MASLD regardless of obesity or IR [105]. Furthermore, it was found that sarcopenic individuals with MASLD are at an increased risk of advanced fibrosis, regardless of their obesity status, IR, or liver enzyme levels [105]. A 1% increase in SMI can lower the risk of MASLD by 20% in men with type 2 diabetes, and handgrip strength has been inversely associated with the presence of MASLD [106,107,108].

### 6.1. Aerobic Exercise

Aerobic exercise, such as moderate-intensity continuous running, cycling, or swimming, has emerged as a promising intervention for individuals with MASLD. It is a low-cost and accessible form of exercise that has demonstrated significant positive effects on various metabolic parameters, with certain limitations in patients with cardiovascular disease and the elderly [109]. Several clinical trials have shown that aerobic exercise can reduce body mass, percentage of fat mass, and total fat mass [110,111]. A recent meta-analysis showed improvement in lipid profiles, particularly in reducing LDL cholesterol levels, increasing HDL levels, and reducing TG levels [112]. Furthermore, aerobic exercise has been shown to increase adiponectin levels, which is crucial for mitigating the inflammatory processes associated with obesity and liver disease [113]. 

### 6.2. Resistance Training

Resistance training involves lifting weights or using resistance equipment to target specific muscle groups, typically the upper arms, abdomen, and legs. Although it may require special equipment and supervision, resistance training has improved metabolic parameters with less energy consumption, making it a safe option for individuals with poor cardiopulmonary fitness [114]. Interestingly, it has a comparable effect to aerobic exercise in slowing the progression of MASLD but with lower energy consumption [115]. One notable outcome of resistance training is enhancing muscle mass, endurance, and strength. Importantly, muscle mass has been shown to stimulate the release of interleukin-6 (IL-6), which directly impacts glucose and lipid metabolism [116]. These findings underscore the potential of resistance training in improving insulin sensitivity and lipid profiles among individuals with MASLD.

### 6.3. Synergistic Effects of Aerobic and Resistance Exercises

Both aerobic and resistance exercises offer individual benefits. One meta-analysis demonstrates the superiority of resistance exercise over aerobic exercise in effectively reducing HFC and TG, improving metabolic syndrome, cardiovascular risk parameters, and body fat [88]. In contrast, this analysis showed that aerobic exercise was better than resistance exercise in BMI reduction [88]. Combining them in a concurrent exercise program has demonstrated superior outcomes in terms of greater improvement in anthropometric measures, a more effective reduction in LDL levels, and a more prominent increase in adipokines levels for individuals with MASLD [116]. A recent meta-analysis of randomized controlled trials shows that the synergistic effects of aerobic and resistance exercise resulted in greater reductions in BMI, both in percentage and kilograms, and an increase in lean body mass, essential for metabolic health. Significant improvement in lipid profiles, especially in LDL cholesterol level, was noted with combined aerobic training and exercise training as opposed to aerobic or resistance training alone [116]. 

### 6.4. Impact of Exercise on Liver Fat and Insulin Sensitivity

All types of exercise have been shown to reduce liver fat content when lasting more than 20 weeks [117]. Also, prolonged exercise interventions of more than 24 weeks provided better health benefits, including improvement in anthropometric indicators of adiposity such as body mass index and fat percentage and mass and lean mass. Combining exercise training has been shown to enhance the benefits by significantly reducing abdominal fat content [88,116].

One systematic review and meta-analysis suggests that exercise training improves glucose disposal compared to individuals not engaging in structured exercise. The study revealed that improvement of insulin sensitivity occurs independently of weight loss, although weight loss can further enhance the effect. However, there was no observed improvement in hepatic insulin sensitivity (related to the liver). The positive effects on insulin action were consistent across age groups (including those over 60 years) and individuals with type 2 diabetes. However, the modality of exercise, particularly aerobic exercise, was a significant predictor of improved glucose disposal. The research found that the cumulative effect of exercise was not dependent on exercise intensity or session length. Adipose tissue insulin resistance index documented a decrease with aerobic exercise training but was not altered by exercise intensity. Interestingly, glucose disposal was enhanced when exercise interventions resulted in significant weight loss. The analysis indicated that every 5% weight loss was associated with improved insulin sensitivity. It was noted that the effects of weight loss on different organ systems vary depending on the extent and duration of metabolic dysregulation. The effects of exercise on glucose disposal primarily stem from skeletal muscle, and exercise did not significantly impact the suppression of hepatic glucose production [118].

### 6.5. Combined Effects of Diet and Exercise

Lifestyle interventions are of paramount importance in the management of MASLD. Clinical trials highlight the combined benefits of diet and exercise, leading to pronounced improvements in weight management, metabolic indicators, and liver-specific parameters [119]. Although diet and exercise alone have exhibited favorable outcomes, combination therapy offers a more potent solution for the challenges of MASLD. 

Current literature reinforces the individual merits of aerobic exercise and diet in diminishing alanine aminotransferase (ALT) levels. One study showed that exercise significantly reduced ALT levels compared to standard physical activity and standard care (−17.55 U/L and −4.94 U/L, respectively) (Figure 5) [120]. Yet, integrating diet and exercise was markedly more effective in decreasing ALT levels (MD, −9.63 IU/L) than exercise alone (MD, −7.59 IU/L) [117]. A subsequent meta-analysis affirmed that a combination of diet and exercise significantly lowered ALT (mean decrease of 14.15 U/L) and AST (mean decrease of 7.33 U/L) relative to the control group [120]. 

Another study revealed that aerobic exercise led to a notable decline in intrahepatic fat (IHF) (MD, −2.72%). This reduction was further amplified when dietary modifications were introduced (MD, −6.61%) [117]. Another investigation revealed that while exercise alone contributed to diminished insulin resistance and bolstered insulin sensitivity, a combination of diet and exercise substantially reduced insulin resistance, as indexed by HOMA–IR scores (MD, −1.99), with an average reduction of 2.07 [120]. With regards to lipid profiles, interventions solely focusing on exercise yielded results akin to combined efforts, elevating HDL levels while reducing triglyceride and LDL concentrations [117,120].

Regarding body weight, an aggregated analysis underscored that merging diet with exercise significantly reduced body weight compared to merely educating about a healthy lifestyle (average decline of 2.82 kg). Although standalone exercise did not markedly decrease weight against the primary control group, it exhibited pronounced reductions compared to routine physical activities and lifestyle education (average weight loss of 8.06 kg and 1.85 kg, respectively) [117,120]. Merging dietary adjustments with exercise results in substantial improvements across diverse health markers for patients with MASLD.

## 7. Role of Gut Microbiota

The human gut microbiota exhibits a pronounced link to the liver, epitomized by the “gut–liver axis” [121]. Dysbiosis, or imbalances in gut microbiota, can engender disturbances in immune function, leading to liver disease. Given that the liver processes signals from the gut in the guise of bacterial products, toxins, and antigens, it crucially maintains a balance between immunity and tolerance, vital for its function [122]. Therefore, the interplay between MASLD and gut microbiota has been scrutinized through probiotics. Probiotics, beneficial microorganisms influencing the host by modulating intestinal microbiota and curbing intestinal inflammation, have been examined in several randomized controlled trials [123,124,125]. Their collective analysis has revealed improvements in hepatic inflammation, demonstrated by decreased ALT, AST, and GGT levels. Additionally, there was a marked reduction in lipid profiles, including TG, LDL-C, and HDL-C, and notable decreases in insulin, insulin resistance, BMI, TNF-alpha, and CRP levels. These findings highlight a potentially important role of probiotics in the management of MASLD [126]. In addition to this, recently, the combination of exercise and probiotics has also yielded effective improvement in the metabolism of MASLD patients, thus reaffirming the importance of probiotics [127,128]. 

## 8. Psychological Aspects and Compliance

The correlation between body weight and MASLD emphasizes the critical need to target obesity as a foundational element in MASLD treatment strategies [129]. The primary management strategy for MASLD hinges on lifestyle modifications, encompassing dietary adjustments and increased physical activity to foster weight loss [130].

Although intensive lifestyle modifications can aid weight loss, these strategies often demand considerable time and resources. One of the main challenges in MASLD remains the enduring alterations to lifestyle modifications. Guiding patients towards sustainable changes, resulting in weight loss and maintaining an optimal weight, can significantly enhance liver health [130]. To mitigate the risk of non-compliance and re-weight gain, it is essential to develop robust patient support systems. These systems should include regular follow-ups, patient education, and motivational interviewing, which are pivotal in fostering lasting lifestyle changes. Furthermore, understanding the behavioral underpinnings that contribute to non-adherence can inform personalized intervention strategies, therefore enhancing the efficacy of MASLD treatment protocols.

Numerous psychological determinants influence eating patterns. Evidence suggests the idea that one’s body weight is not exclusively a result of dietary choices. Instead, it reflects their distinctive cognitive, emotional, and behavioral characteristics [131]. It is crucial to identify the psychological and behavioral elements that encourage patients to sustain their lifestyle modifications [129].

The trans-theoretical model of behavior change posits that individuals navigate various stages while endeavoring to alter their behavior: pre-contemplation, contemplation, preparation, action, and maintenance. Progression through these stages is non-linear r; individuals might oscillate between stages as they pursue behavioral change. Motivational interviewing (MI) is a clinical technique devised to aid patients in discerning their intrinsic motivation for positive changes and tackling their ambivalence. MI elucidates that traditional medical interactions, characterized by a top-down flow of advice, might render some patients even more resistant to change [129]. In healthcare, an emphasis on understanding the patient’s vantage point is indispensable. Rather than directive communication, clinicians should prioritize dialogues, aiming to comprehend patient experiences. Such a collaborative approach enhances the shared decision-making process concerning treatment pathways [130].

MI’s efficacy in augmenting weight loss outcomes within structured weight management frameworks has been corroborated [129]. However, research elucidating the adherence to exercise regimens post-MI still needs to be explored across diverse scenarios and populations [132].

An evolution towards a collaborative team-based methodology is essential for the holistic treatment of MASLD. Such an approach mandates the union of medical and behavioral faculties, operating synergistically to deliver tailored care [129].

## 9. Special Populations

### 9.1. Prediabetics

Prediabetes represents an intermediate stage where blood sugar levels are elevated (fasting plasma glucose levels from 100 to 125 mg/dL) but have not yet reached the diabetic threshold. It shares underlying mechanisms with diabetes, but the degree of insulin resistance is less pronounced. Importantly, prediabetes can potentially be reversed through lifestyle changes [133]. It is typically characterized by HbA1c levels between 5.7% and 6.4% [133].

A pivotal factor in the development of MASLD is insulin resistance. This condition arises from enhanced gluconeogenesis and glycogenolysis in the liver, reduced glucose uptake in peripheral muscles leading to hyperglycemia, and increased insulin levels. Additionally, the upregulation of lipogenic factors and heightened hepatic lipogenesis further drive the progression of MASLD [134]. MASLD is a robust and independent predictor for prediabetes in the general adult populace. Hence, patients with MASLD should be recognized as a high-risk group. 

### 9.2. Children

Children with biopsy-proven hepatic steatosis are more likely to develop prediabetes or type 2 diabetes than their counterparts without MASLD [135]. Notably, even after accounting for age and BMI, girls with MASLD appear more predisposed to type 2 diabetes than boys with the same condition [136]. Adults with MASLD also commonly exhibit abnormal glucose regulation [136]. Regardless of age, individuals with MASLD and abnormal glucose tolerance are more likely to progress to metabolic dysfunction-associated steatohepatitis (MASH) than those with standard glucose tolerance [135,136,137]. 

### 9.3. Advanced Age

As individuals advance in age, the elderly undergo physiological changes, leading to functional decline and frailty [138]. MASLD is prevalent among the elderly, and its liver and non-liver-related complications are more pronounced than in younger individuals [139]. As MASLD is associated with numerous comorbidities in the elderly, intervention is imperative. However, recommended lifestyle modifications, such as diet and exercise, might be met with skepticism, particularly concerning the feasibility of increased physical activity. Furthermore, when contemplating pharmacological treatments, it is crucial to balance risks and benefits, especially given the high incidence of comorbidities and polypharmacy in the elderly [139].

## 10. Future Directions

Prior research on diet and exercise in the management of MASLD showed that lifestyle changes were thought to be the most integral in altering the progression of the disease. Insulin resistance and altered levels of glucose and lipid metabolism play a role in the progression of MASLD. There are many ongoing clinical trials for future treatment, including ones that target these modes of progression. Currently, none are approved. Reprogramming anti-diabetes and anti-obesity medication is a major treatment option. Multiple pathways involving hepatic inflammation, fibrosis, and cell death are critically important targets for therapy [140].

There are many pathways to target when treating MASLD. TGF-B1 plays a key role in liver cell apoptosis, fibrosis, and liver inflammation. Peroxisome proliferator-activated receptors are nuclear protein receptors that play a vital role in modulating fatty acid. There are numerous other options when choosing to treat MASLD. Glucose metabolism, Kruppel-like factors, insulin signaling pathways, wnt signaling pathways, p53 pathway with the use of huh7 cells, vascular cell adhesion molecule 1, GLP-1 receptors [141].

Currently, PPAR has been used for T2DM, hyperlipidemia, metabolic syndrome, and cardiovascular disease. Rosiglitazone, a medication used for type 2 diabetes, has shown ample effects against steatosis, hepatocellular inflammation, ballooning degeneration, and fibrosis. A recent clinical trial showed that lobeglitazone decreased steatosis and glycemic control and controlled liver enzymes for liver damage [140]. Saroglitazar, another PPAR agonist, can be used to reverse the progression of MASLD. In a double-blind study, 4 mg of saroglitazar resulted in a significant reduction of ALT and LFCs compared to placebo at week 4 and 16-week intervals. Additionally, markers of hepatocellular injury and fibrosis, including CK18, ELF, LSM, and APRI, showed significant improvement following 16 weeks of drug use. In a dose-dependent fashion, saroglitazar decreased VLDL-TG, LDL-TG, HDL-TG/HDL-C ratio, along with decreasing the levels of several bile acids, which play a key role in MASLD progression [142].

MiRNAs have roles in the early progression and development of MASLD. miR-21 can help obesogenic diet-induced steatosis and glucose intolerance as seen in multiple studies with mice. Different miRNAs can modulate different pathways and help to decrease steatosis and fibrosis in many patients. miRNA can be used as an early noninvasive marker for early MASLD before progressing to HCC, cirrhosis, or fibrosis. miRNA can be used additionally as a potentially noninvasive diagnostic marker for the progression of MASLD. Early diagnosis of MASLD is key to preventing later irreversible fibrosis, cirrhosis, and HCC. Once diagnosed using the marker, many modalities, such as PPAR, GLP-1, and KLF, can be used to treat and prevent progression [141].

In current studies, it is still unclear if Vitamin E is beneficial long-term in the progression of MASLD. A recent meta-analysis reports nine randomized control trials involving the efficacy of Vitamin E in current treatments [143]. Five were RCTs with adults, and 4 were done on children. Four out of five studies showed improvement in ALT levels. On the other hand, only one of four RCTs showed Vitamin E as beneficial for children in improving ALT levels. The study showed biochemical and histological improvement in adults with Vitamin E as a treatment in the progression of MASLD. The study was limited in length as more time would be needed to see if Vitamin E provides a long-term benefit [143].

Another study showed that Vitamin E was beneficial in the long and short term for adult and pediatric patients, decreasing ALT and AST levels and slowing the progression of MASLD [144]. The dose-dependent studies ranging from 1 dose to 1000 IU showed a significant reduction in AST and ALT levels at a 6-month follow-up and pediatric liver function levels at a 12-month follow-up. Fibrosis score and fibrosis levels significantly dropped after 24 months. The meta-analysis showed that for pediatric patients, the long-term effects of Vitamin E were most beneficial, and for adult populations, the most promising patients were obese and between ages 18–50 with baseline AST over 50 and a dose of 400–800 IU who are actively trying to lose weight [144].

The role of bariatric surgery in MASLD has rapidly expanded over the past decade. Roux-en-Y gastric bypass (RYGB) and sleeve gastrectomy are the two most common bariatric procedures worldwide. Unique modalities that are under study as additional options include an intragastric balloon in patients with MASLD and advanced fibrosis, which have shown a reduction in liver stiffness and FIB-4. IBG therapy was placed in obese patients without signs of endoscopic portal hypertension for 6 months [145]. This retrospective analysis showed that 6 months of silicon-made IGB can induce weight loss in obese patients and improve fibrosis by decreasing liver stress. This modality is limited due to side effects, including nausea and reflux symptoms, causing many patients to request removal after some time despite anti-emetics and PPI treatments [145].

Duodenal mucosal resurfacing (DMR) is another novel endoscopic procedure aimed at achieving metabolic effects by resurfacing the duodenum. It is hypothesized that mucosal remodeling may reset duodenal enteroendocrine cells that have become diseased [146]. This is carried out endoscopically by placing a balloon into the duodenum and ablating the submucosa with cold and hot water. The goal is to complete circumferential ablation of the postpapillary duodenum. The first human study decreased A1C levels by 1.2% in 39 patients at 6 months, with patients with a baseline A1C of 9.6%. In the European cohort (DMR: *n* = 39; sham: *n* = 36), the modified intention-to-treat analysis showed that the median HbA1c at 24 weeks post-procedure decreased by −6.6 (17.5) mmol/mol in the DMR group compared with −3.3 (10.9) mmol/mol in the sham procedure group (*p* = 0.033). In patients with baseline liver magnetic resonance imaging proton density fat fraction >5% and baseline fasting plasma glucose (FPG) ≥10 mmol/L, there was an absolute reduction of liver fat content by 7.6% in the DMR group compared with a 3.1% reduction in the sham procedure group at 12 weeks (*p* = 0.001). DMR is a novel procedure that has very beneficial effects on T2DM patients, but the studies regarding MASLD progression are limited but promising [147].

A biliopancreatic diversion with a duodenal switch is a variant of BPD in which sleeve gastrectomy (SG) is performed instead of horizontal gastrectomy. Patients who underwent DS-SG alone were compared with those who received SG alone and the combination. At 1 year, the DS-SG group experienced 39 ± 13% EWL compared with 47 ± 19% EWL in the SG group. The HbA1c level decreased by 10% and 19% in the SG and DS-SG groups, respectively [146]. Another option is the “Incisionless Magnetic Anastomotic System”. The IMAS (GI Windows) creates an anastomosis via incisionless magnetic compression. The HbA1c level decreased from 6.6 ± 1.8% to 5.4 ± 0.5% [78]. The postprandial GLP-1 level increased at 2 months; however, longer-term results are needed to draw further conclusions [146].

In the future, it would be beneficial to do a retrospective study combining the effect of physical activity and a compound that would block the pathway of progression for fibrosis or even in combination with an endoscopic procedure. The outlook is promising regarding the multiple treatment options. 

## 11. Conclusions

This comprehensive review of MASLD highlights the pathophysiology of the disease along with the lifestyle modifications, pharmacology, and endoscopic/surgical modalities that can be of benefit. With regard to dietary modifications, intermittent fasting, the Mediterranean diet, low-carb/low-fat diet, high-protein diet, and ketogenic diet offer distinctive benefits for MASLD patients. Exercise, spanning aerobic, resistance, and combined forms, is an effective non-pharmacological option, substantially reducing liver fat, enhancing insulin sensitivity, and mitigating other metabolic risk factors.

Clinically, the integrated effects of diet and exercise surpass singular interventions. Notably, combined diet and exercise regimens, such as calorie-restricted Mediterranean diets supplemented with physical activity, showcase pronounced reductions in liver-specific parameters and metabolic indicators. Moreover, the gut–liver axis and the potential of probiotics emerge as new frontiers in clinical therapy, offering avenues to combat inflammation and enhance liver health. Additionally, more effective pharmacologic options may be available on the horizon for MASLD. Bariatric surgery may have benefits in MASLD, and the emergence of endoscopic sleeve gastroplasty may present another viable option for patients. Psychological perspectives highlight the importance of patient engagement and motivation, with models such as motivational interviewing offering promise in ensuring adherence to lifestyle changes.

## Figures and Tables

**Figure 1 jpm-14-00061-f001:**
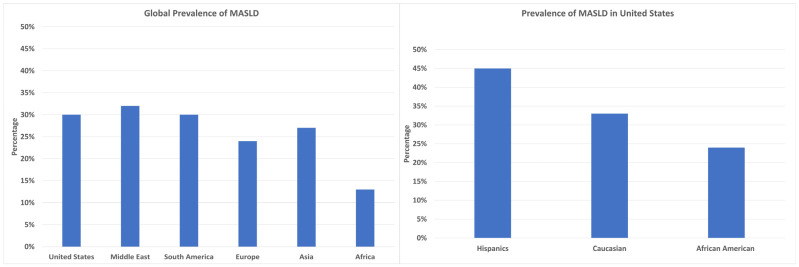
Prevalence of metabolic-associated fatty liver disease (MASLD) globally and in the United States. Percentages reported by previous studies as cited above.

**Figure 2 jpm-14-00061-f002:**
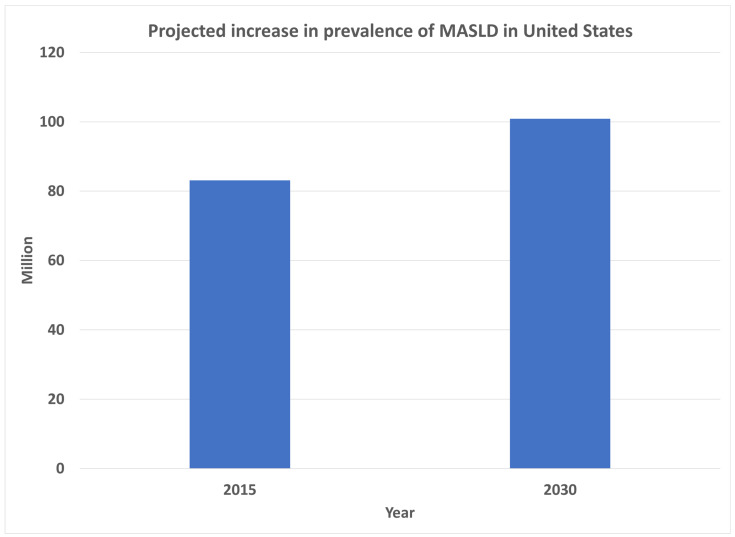
Projected rates of MASLD as reported by Friedman SL et al. [22].

**Figure 3 jpm-14-00061-f003:**
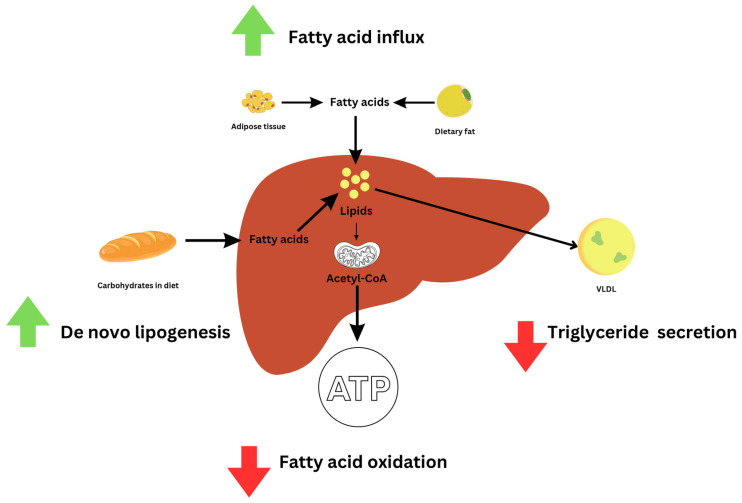
Summary of the primary routes leading to triglyceride buildup in MASLD. ”VLDL” and “ATP”.

**Figure 4 jpm-14-00061-f004:**
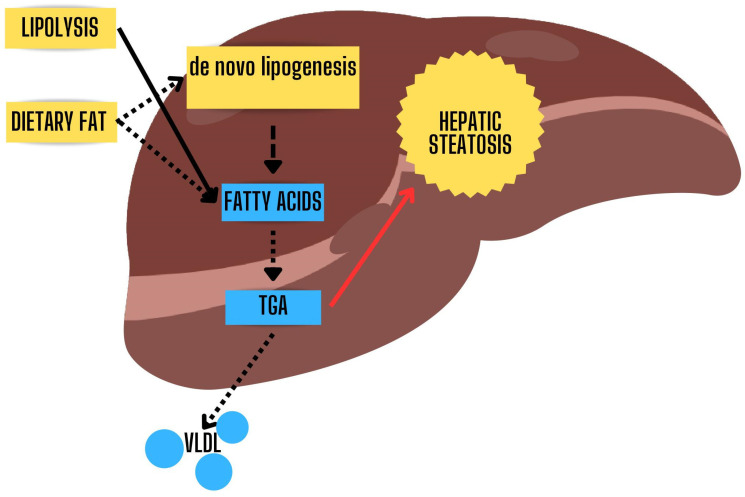
Role of de novo lipogenesis in hepatic steatosis. TGA: Triglyceride.

**Figure 5 jpm-14-00061-f005:**
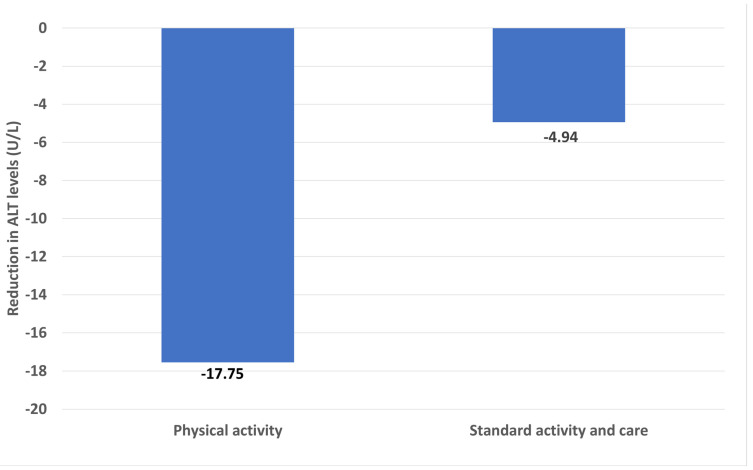
Reduction in Alanine transaminase levels (ALT) secondary to exercise as reported by Fernández et al. [107].

## Data Availability

This review article has no associated data.

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
