# Peer review of "Non-Pharmacological Approach to Diet and Exercise in Metabolic-Associated Fatty Liver Disease: Bridging the Gap between Research and Clinical Practice"

_jpm, 2024, doi:10.3390/jpm14010061_

Round 1

Reviewer 1 Report

Comments and Suggestions for Authors

I read the article with great interest. Nevertheless, many points should be clarified.

  • In the introduction, you should briefly describe clinical and histological differences between different nomenclatures of fatty liver disease (NAFLD, MAFLD, MASLD, and MetALD) (see here: doi: 10.1016/j.jhep.2023.08.021). Then, why should the term MASLD be prefered (doi: 10.1016/j.jhep.2023.07.031).
  • In describing the progression of MASLD, you should mention the role of mithocondria and dysfunction linked to liver damage. Many strategies to prevent or restore liver function focus on the improvement of mitochondrial activities (see doi: 10.1053/j.gastro.2018.06.083, doi.org/10.1016/j.molmet.2020.101134).
  • Regarding the importance of diet in MASLD (chapter 5), a general overview of the diet approach to improving NAFLD is missing (see DOI: 10.1111/liv.15024, DOI: 10.3390/cells10071805). However, before talking about a specific diet, remember that weight loss generally has a positive impact on liver health rather than a particular kind of diet. Then, paragraph 5.4 should be eliminated, mainly because defining a diet based on a percentage of protein is wrong. However, be careful with the proper definition of ketogenic diet (doi: 10.1017/S0007114521002609). The role of a ketogenic diet has been suggested in reducing oxidative stress (doi: 10.3390/antiox12051065) and supporting muscle mass (doi: 10.3390/nu14030620).
  • Regarding the importance of exercise (chapter 6), more attention should be dedicated to the role of muscle in global health (org/10.1210/endrev/bnaa016), considering also that a lower muscle mass is related to a higher risk of liver disease (doi: 10.1155/2017/6297651, doi: 10.1007/s12072-019-09996-7).

chapter 7: “Combined Effects of Diet and Exercise,” so in the previous paragraph, you referred to only exercise interventions? Generally, this section, together with the role of exercise, should be rewritten.

Why do you consider prediabetics, children, and older adults as a special population?

Finally, as you claim in the title, in what way is your article bridging the gap between research and clinical practice?

Author Response

Reviewer 1:

I read the article with great interest. Nevertheless, many points should be clarified.

In the introduction, you should briefly describe clinical and histological differences between different nomenclatures of fatty liver disease (NAFLD, MAFLD, MASLD, and MetALD) (see here: doi: 10.1016/j.jhep.2023.08.021). Then, why should the term MASLD be prefered (doi: 10.1016/j.jhep.2023.07.031).

Author response:  The reviewers pointed out a very important point which should be mentioned in the introduction. The authors have included all the definitions which are included in Line 47-55.

“In early 2020, a group of international experts led a consensus-driven effort to create a more suitable name for the disease suggested renaming NAFLD as metabolic dys-function-associated fatty liver disease (MAFLD). [2,3] Three years later, the term metabolic dysfunction-associated steatotic liver disease (MASLD) has been proposed, and it can be diagnosed based on a patient meeting one of five cardiovascular risk factors, unlike MAFLD, which required that patients meet two of seven parameters of metabolic dysfunction. Among them, patients who meet both MASLD and alco-hol-related fatty liver disease (ALD) criteria are categorized as having MetALD. [4,5]”

In describing the progression of MASLD, you should mention the role of mitochondria and dysfunction linked to liver damage. Many strategies to prevent or restore liver function focus on the improvement of mitochondrial activities (see doi: 10.1053/j.gastro.2018.06.083, doi.org/10.1016/j.molmet.2020.101134).

Author response:  The reviewers mentioned discussing the role of mitochondrial dysfunction which definitely plays a critical role in causing liver damage. The authors have included an additional section in the article which highlights the importance of this topic. Paragraph 4.9 starting from Line 502-527 is included in the revised manuscript.

“Mitochondria play a pivotal role in the regulation of hepatic lipid metabolism and the management of oxidative stress. In liver tissues from patients with both alcohol-related and non-related liver disorders, there are observable changes such as ultrastructural damage to mitochondria, altered mitochondrial dynamics, reduced respiratory chain complex activities, and a compromised capacity for adenosine triphosphate synthesis. [67] The balance shifts towards increased lipogenesis and reduced fatty acid β-oxidation, resulting in triglyceride accumulation within hepatocytes. This imbalance, along with elevated reactive oxygen species, contributes to insulin resistance in steato-hepatitis patients. Mitochondrial reactive oxygen species are also key in signaling metabolic path-ways, and any disruption in these pathways can influence the onset and progression of chronic liver diseases. The stress and damage to mitochondria are im-plicated in cellular death, the fibrogenesis of the liver, inflammation, and innate immune responses to viral infections. Therefore, mitochondrial functions are entwined with the development of various chronic liver conditions, such as nonalcoholic fatty liver disease, alcohol-associated liver disease, drug-induced liver injury, and viral hepatitis B and C. This exposure hints the potential therapeutic strategies targeting these mitochondrial processes. [67] The equilibrium between the oxidation and storage of fat hinges on the type of fuel mitochondria utilize, rather than their capacity to generate ATP through respiration. Therefore, therapeutic approaches that modulate mitochondrial fuel choice could be more effective for managing non-alcoholic fatty liver disease (NAFLD). In parallel, it may be more beneficial to inhibit maladaptive antioxidant responses instead of disrupting the normal mitochondrial hydrogen peroxide-driven signaling pathways, preserving proper hepatic insulin signaling in NAFLD. Investigating the specific roles of different antioxidant systems within subcellular compartments, as well as the distinct roles played by various mitochondrial subpopulations, could unveil novel targets for NAFLD treatment. [68]”

Regarding the importance of diet in MASLD (chapter 5), a general overview of the diet approach to improving NAFLD is missing (see DOI: 10.1111/liv.15024, DOI: 10.3390/cells10071805). However, before talking about a specific diet, remember that weight loss generally has a positive impact on liver health rather than a particular kind of diet. Then, paragraph 5.4 should be eliminated, mainly because defining a diet based on a percentage of protein is wrong. However, be careful with the proper definition of ketogenic diet (doi: 10.1017/S0007114521002609). The role of a ketogenic diet has been suggested in reducing oxidative stress (doi: 10.3390/antiox12051065) and supporting muscle mass (doi: 10.3390/nu14030620).

Author response:  The authors have included a general overview of the diet approach in the article as aptly pointed out by the reviewers. Changes made in Line 529-542.

Paragraph 5.4 has been eliminated as suggested by the reviewers and authors agree with the suggestions. Line 586-593 have been deleted.

The definitions of the ketogenic diet have been updated appropriately with the explanation of the role of ketogenic diet in reducing oxidative stress reflected in Line 595-607

Although no medications are currently approved specifically for the treatment of metabolic-associated fatty liver disease (MASLD) and related cirrhosis, lifestyle modifications including diet and physical activity are widely accepted as foundational treatments for NAFLD/NASH. 69 While recognized as essential for addressing the NAFLD epidemic, existing guidelines offer imprecise and broad directives for dietary and exercise interventions for affected individuals. 70

Several scientific associations (EASL‐EASD–EASO 2016, 71 AASLD 2018, 72 ESPEN 2019 73 and APASL 2020 74) emphasize the significance of weight loss—aiming for a 7-10% reduction in body weight achieved by a hypocaloric diet (energy deficit of 500‐1000 kcal/d) and/or PA (in order to promote a caloric deficit). Despite the consensus on the objective of weight loss, the specifics of the recommendations vary across different associations.

Weight loss is crucial in reversing MASLD. We discuss different dietary approaches that target weight loss and improve liver health through a variety of mechanisms.”

“Ketogenic diet:

MAFLD is associated with disrupted lipid metabolism, often due to mitochondrial dys-function which initiates a harmful cycle that exacerbates oxidative stress and triggers inflammation, leading to the progressive loss of hepatocytes and advancing MAFLD to its more severe stages. A ketogenic diet, characterized by very low carbohydrate intake (less than 30 grams per day) that leads to "physiological ketosis," has shown promise in mitigating oxidative stress and improving mitochondrial function. [90] There has been evidence provided by studies that shows that the ketogenic diet (KD) can enhance metabolic health and increase the population of γδ T cells within adipose tissue, which play a critical role in controlling blood sugar levels in the context of obesity. Consequently, the KD is being considered as a potential treatment for individuals with sarcopenic obesity, due to its beneficial impacts on visceral adipose tissue (VAT), adipose tissue regulation, inflammatory markers including cytokines, blood biochemistry, gut microbiota, and overall body composition. [91]”

Regarding the importance of exercise (chapter 6), more attention should be dedicated to the role of muscle in global health (org/10.1210/endrev/bnaa016), considering also that a lower muscle mass is related to a higher risk of liver disease (doi: 10.1155/2017/6297651, doi: 10.1007/s12072-019-09996-7).

Author response:  As reviewer’s aptly mentioned the importance of discussing the role of muscle in global health and relation of lower muscle mass related to a higher risk of liver disease, the authors have made the appropriate changes in Line 635-647.

“Diminished muscle mass is linked with lower survival rates, prolonged hospital stays, and increased mortality in cirrhotic patients. [103] Muscle function also has a reciprocal relationship with non-alcoholic fatty liver disease (NAFLD). Studies have shown that individuals with reduced muscle mass are at a heightened risk of developing NAFLD, even when factors like insulin resistance (IR) and inflammation are accounted for. [104] Skeletal muscle index (SMI) has a converse relationship with markers such as HOMA-IR, hs-CRP, triglycerides, and overall body fat percentage NAFLD even after adjusting for potential confounding factors. [104] Another subsequent study showed a positive relationship between sarcopenia and NAFLD regardless of obesity or IR. [105] Furthermore, it was found that sarcopenic individuals with NAFLD are at an increased risk for advanced fibrosis, regardless of their obesity status, IR, or liver enzyme levels. 105A 1% increase in SMI can lower the risk of NAFLD by 20% in men with type 2 diabetes, and handgrip strength has been inversely associated with the presence of NAFLD. [106,107]”

chapter 7: “Combined Effects of Diet and Exercise,” so in the previous paragraph, you referred to only exercise interventions? Generally, this section, together with the role of exercise, should be rewritten.

Author response: The author’s have moved this paragraph with in section 6 and deleted it from section 7 to focus on the combined effects of diet and exercise.

Why do you consider prediabetics, children, and older adults as a special population?

Author response: Yes, the author would like to consider prediabetics, children, and older adults are considered special populations in the context of metabolic (dysfunction) associated fatty liver disease (MAFLD) due to their unique physiological and metabolic characteristics. For prediabetics, the state of insulin resistance makes them particularly susceptible to the development of MAFLD. Their metabolic profile is often accompanied by other metabolic syndrome components such as hypertension and dyslipidemia, can exacerbate the progression of liver disease. Children represent a special population because the etiology, progression, and management of MAFLD can be different in this group compared to adults. The impact of MAFLD on growing bodies, potential long-term effects, and the role of pediatric obesity are of particular concern. Additionally, the diagnosis and treatment strategies in children must be tailored to their developmental stage and consider the potential impact on growth and development. Older adults are considered a special population because they often have comorbid conditions and age-related changes in body composition, such as sarcopenia, which can influence the presentation and progression of MAFLD. Furthermore, the presence of multiple medications due to comorbidities can affect liver function and complicate the management of MAFLD. In this population, the risk-benefit ratio of interventions might also differ due to the presence of frailty and a potentially limited life expectancy.

Finally, as you claim in the title, in what way is your article bridging the gap between research and clinical practice?

Author response: The article bridges the gap between research and clinical practice by translating evidence-based non-pharmacological interventions for Metabolic Associated Fatty Liver Disease (MASLD) into practical recommendations. It does so by outlining the benefits of specific dietary patterns and physical exercise regimens that have been shown to be effective in improving metabolic health and liver function. This includes the impact of coffee consumption, intermittent fasting, Mediterranean and ketogenic diets, and the combination of aerobic and resistance training.

By emphasizing an integrated approach that combines diet and exercise with gut microbiota considerations, the review goes beyond the theoretical to address real-world clinical application. It stresses the importance of tailoring interventions to individual patient needs and motivations, thus advocating for a personalized treatment strategy that is essential in clinical settings.

Moreover, the review contrasts lifestyle interventions with pharmacological and surgical options, highlighting their long-term benefits and lower risk profiles. This comparison serves to inform clinicians about the relative advantages of lifestyle changes as a first-line intervention.

The call for ongoing patient education and continued research underscores the dynamic nature of clinical practice, where new findings constantly inform and improve patient care. Overall, the article serves as a conduit for applying current research to patient management, aligning with the evolving landscape of personalized medicine in the management of MASLD.

Reviewer 2 Report

Comments and Suggestions for Authors

I feel that you have accomplished a fairly solid review of the subject matter but have not brought a significant amount of new information to the table.  

I also feel that you have left out one key aspect of the treatment using non-pharmacological methods for the treatment of MASLD - how to maintain the long term diet and exercise. You did touch on it in section 9, but I wonder if you might not have found any more information relating to how non-compliance and/or re-weight gain would affect MASLD. This could be important as well. Incorporating aspects of treating behavioral issues will be just as important as diet and exercise.

As for the manuscript itself:

Overall, it was well written with the exception of the issues noted below in the language section.

I did note that there were many abbreviations used throughout the manuscript that got quite confusing.  Also, there were numerous instances where they were not defined on first usage.  I would also suggest a list of the abbreviations for quick referral. 

Line 160, you noted a study of 100 subjects with Type 2 DM of which 49% had hepatic steatosis and stated that this "confirmed" as strong independent risk factor for MASLD.  Is this really a good confirmation? I submit that the study population is too small to be a "confirmation".  You would need a much larger study over a much larger geographical area to say that.  Consider re-wording.

Comments on the Quality of English Language

Overall, I found the manuscript understandable however there were many instances particularly within the first half of the manuscript in which it appeared that English may not be the writer's primary language as there were phrasing issues and incomplete sentences.  

I would suggest that either a native English speaking person or an English writing service review the manuscript prior to re-submission.

Some examples to look at:

Lines:

84; 120-121; 131-133; 199; 201 and there are more

Author Response

Reviewer 2

I feel that you have accomplished a fairly solid review of the subject matter but have not brought a significant amount of new information to the table. 

I also feel that you have left out one key aspect of the treatment using non-pharmacological methods for the treatment of MASLD - how to maintain the long term diet and exercise. You did touch on it in section 9, but I wonder if you might not have found any more information relating to how non-compliance and/or re-weight gain would affect MASLD. This could be important as well. Incorporating aspects of treating behavioral issues will be just as important as diet and exercise.

Author response: Thank you for your insightful comments. We appreciate the opportunity to enhance our manuscript by expanding on the long-term adherence to non-pharmacological interventions in MASLD. We agree that patient compliance and behavioral aspects are critical for sustained treatment success, this has now been mentioned in our manuscript under section 9.

Overall, it was well written with the exception of the issues noted below in the language section.

I did note that there were many abbreviations used throughout the manuscript that got quite confusing.  Also, there were numerous instances where they were not defined on first usage.  I would also suggest a list of the abbreviations for quick referral.

Author response: We acknowledge that the frequent use of abbreviations might have caused confusion. To address this, we will ensure that all abbreviations are defined upon their first use.

Line 160, you noted a study of 100 subjects with Type 2 DM of which 49% had hepatic steatosis and stated that this "confirmed" as strong independent risk factor for MASLD.  Is this really a good confirmation? I submit that the study population is too small to be a "confirmation".  You would need a much larger study over a much larger geographical area to say that.  Consider re-wording.

Author response: We appreciate your suggestion for caution in presenting our findings. The wording 'confirmed' has been reconsidered and changed.

Overall, I found the manuscript understandable however there were many instances particularly within the first half of the manuscript in which it appeared that English may not be the writer's primary language as there were phrasing issues and incomplete sentences.  I would suggest that either a native English speaking person or an English writing service review the manuscript prior to re-submission.

Some examples to look at:

Lines: 84; 120-121; 131-133; 199; 201 and there are more

Author response: We are grateful for your recommendations concerning the language used in our manuscript. We have reviewed the manuscript thoroughly to rectify any phrasing issues and incomplete sentences, particularly in the specified lines. Majority of the author’s native language is in English, although they do not have predominant Caucasian names.